# Coronavirus Inhibitors Targeting nsp16

**DOI:** 10.3390/molecules28030988

**Published:** 2023-01-18

**Authors:** Ejlal A. Omer, Sara Abdelfatah, Max Riedl, Christian Meesters, Andreas Hildebrandt, Thomas Efferth

**Affiliations:** 1Department of Pharmaceutical Biology, Institute of Pharmaceutical and Biomedical Sciences, Johannes Gutenberg University, Staudinger Weg 5, 55128 Mainz, Germany; 2Institute for Medical Informatics, Statistics and Epidemiology, University of Leipzig, 04107 Leipzig, Germany; 3High Performance Computing Group, University of Mainz, 55131 Mainz, Germany; 4Institute for Computer Science, University of Mainz, 55131 Mainz, Germany

**Keywords:** SARS-CoV-2, SARS-CoV-1, MERS-CoV, nsp16, natural products, pan-inhibitor, virtual drug screening

## Abstract

During the past three decades, humans have been confronted with different new coronavirus outbreaks. Since the end of the year 2019, COVID-19 threatens the world as a rapidly spreading infectious disease. For this work, we targeted the non-structural protein 16 (nsp16) as a key protein of SARS-CoV-2, SARS-CoV-1 and MERS-CoV to develop broad-spectrum inhibitors of nsp16. Computational methods were used to filter candidates from a natural product-based library of 224,205 compounds obtained from the ZINC database. The binding of the candidates to nsp16 was assessed using virtual screening with VINA LC, and molecular docking with AutoDock 4.2.6. The top 9 compounds were bound to the nsp16 protein of SARS-CoV-2, SARS-CoV-1, and MERS-CoV with the lowest binding energies (LBEs) in the range of −9.0 to −13.0 kcal with VINA LC. The AutoDock-based LBEs for nsp16 of SARS-CoV-2 ranged from −11.42 to −16.11 kcal/mol with predicted inhibition constants (pKi) from 0.002 to 4.51 nM, the natural substrate S-adenosyl methionine (SAM) was used as control. In silico results were verified by microscale thermophoresis as in vitro assay. The candidates were investigated further for their cytotoxicity in normal MRC-5 lung fibroblasts to determine their therapeutic indices. Here, the IC_50_ values of all three compounds were >10 µM. In summary, we identified three novel SARS-CoV-2 inhibitors, two of which showed broad-spectrum activity to nsp16 in SARS-CoV-2, SARS-CoV-1, and MERS-CoV. All three compounds are coumarin derivatives that contain chromen-2-one in their scaffolds.

## 1. Introduction

COVID-19 was detected for the first time in December 2019 in Wuhan, China, where 41 patients were diagnosed with pneumonia of unknown etiology. The symptoms included pharyngitis, dry cough, intermittent fever, tiredness, and dyspnea. Some patients developed respiratory distress syndrome. Therefore, they required ventilatory support. On 7 January 2020, the Chinese Centre for Disease Control and Prevention (CCDC) identified a new β-coronavirus from throat swab samples belonging to the same family as SARS-CoV-1 and MERS-CoV as the causative agent. The newly identified virus was named Severe Acute Respiratory Syndrome Coronavirus 2 (SARS-CoV-2), whilst the disease was called by the World Health Organization (WHO) Coronavirus Disease 2019 (COVID-19) [1]. Within a short time, the disease spread outside China. Consequently, WHO declared it a public health emergency of international concern on January 30th and then a pandemic on 11 March 2020 [2,3]. Within a few months, COVID-19 terrifyingly stroked almost all countries worldwide. To slow down the spreading of the disease, most countries applied measures to reduce pressure on their health systems including strict limitations of social contact, travel restrictions, and—most notably—lockdowns. These applied measures had considerable negative economic and social impacts. Moreover, severe and life-threatening SARS-CoV-2 infections led to the collapse of healthcare systems in many countries [4].

Coronaviruses belong to the family Coronaviridae which infect amphibians, birds, and mammals including humans, camels, pigs, and bats. Coronaviruses are divided into four genera: α-, β-, γ-, and δ-coronaviruses. The 7 identified human coronaviruses are all either α- or β-coronaviruses. Four of the human coronaviruses (229E, NL63, OC43, and HKU1) show seasonal waves during winter causing mild to moderate symptoms of upper respiratory tract infection. They are responsible for common cold symptoms that infect most people repeatedly over their lifetime. The other three viruses SARS-CoV-2, SARS-CoV-1, and the Middle East Respiratory Syndrome Virus (MERS-CoV) cause severe symptoms and higher mortality rates estimated with up to 3.5%, 10%, and 35%, depending on particular strains, respectively [5].

SARS-CoV-2 represents an enveloped, positive-sense, large single-stranded RNA virus (~30 kb). It contains 15 open reading frames (ORFs) encoding a number of structural proteins, non-structural proteins and accessory proteins. The structural proteins include nucleocapsid, membrane, envelope and spike protein. Accessory proteins are orf3a, orf6, orf7a, orf7b, orf8, and orf10. There are 16 non-structural proteins (nsp1–16), these non-structural proteins are encoded by both ORF1a and ORF1b, the former ORF encodes nsp1 to nsp11, the latter one encodes nsp12 to nsp16 [6,7].

Non-structural proteins are essential for viruses. They hold the viral enzymes such as viral proteases and RNA-dependent RNA polymerase which are important for the translation, replication, and proliferation of the virus [8,9].

The virus RNA cap is vital for the stability of RNA. The most remarkable feature of the coronavirus structure is the 5′end RNA cap. It resembles the structure of the mRNA of humans, they both consist of C2′-*O*-methylrybosyladenine and N-methylated guanosine triphosphate. This mimicry in structure enables the virus to use the machinery of the host for protein production as well as evade the immune system of the host [10]. The installation of type 1 cap on RNA includes four steps catalyzed by 5′-RNA triphosphatase, guanylyl transferase, N-7 methyltransferase and 2′-*O*-methyltransferase, and the latest methylation is executed by nsp16.

Nsp16 is an S-adenosylmethionine (SAM)-dependent methyltransferase. It is well-conserved among coronaviruses, and it is essential for coronavirus replication [11,12]. The disruption of the binding between nsp16 and the substrate SAM inhibits its methyltransferase activity and, consequently, inhibits viral replication [13,14,15]. Accordingly, the SAM binding site is considered one of the attractive targets in COVID-19 drug discovery.

Since the beginning of the pandemic, efforts have been continued to develop diagnostic, preventive, and treatment strategies. As a result, a number of antiviral drugs were found to reduce mortality and hospitalization rates. Moreover, vaccines are now available, including mRNA vaccines, adenovirus vectors, inactivated viruses, and subunit vaccines [16]. Still, there is an urgent need for novel drug candidates to combat newly emerging and circulating infections of coronaviruses [17,18]. As the pandemic is global, the need for economically affordable anti-coronaviral agents with convenient dosage forms remains persistent.

The present study aims to discover natural drug candidates utilizing coronaviral nsp16 as a potential druggable target. As previously mentioned, nsp16 is well-conserved among coronaviruses as well as it is vital for virus replication. Therefore, we targeted nsp16 to identify broad-spectrum anti-coronaviral candidates that inhibit this protein in SARS-CoV-2, SARS-CoV-1, and MERS-CoV. According to several surveys by the National Cancer Institute, USA, natural products and their derivatives represent the major source of drugs for many diseases [19]. Therefore, we assume that the identification of chemical scaffolds from natural sources is a good starting point for drug development. Once natural product-based lead compounds are at hand, their pharmacological features can be further improved by synthetic or semisynthetic chemical derivatization. A natural product library derived from the ZINC database was selected for screening as one of the largest databases available at open access databases with commercially available compounds [20]. A total of 224,205 ligands were screened in silico against nsp16 of SARS-CoV-2, SARS-CoV-1, and MERS-CoV. The top 9 candidates were further investigated in vitro. As a result, three ligands were identified as potential SARS-CoV-2 nsp16 inhibitors, and two of them were broad-spectrum inhibitors of nsp16 in all investigated coronaviruses.

## 2. Results

### 2.1. Multiple Sequence Alignment

Multiple sequence alignment was performed to identify the resemblance between the nsp16 proteins of SARS-CoV-2, SARS-CoV-1, and MERS-CoV. The percentage of identity matrix was 92.69% between SARS-CoV-2 and SARS-CoV-1, 66.11% between SARS-CoV-2 and MERS-CoV, and 65.10% between SARS-CoV-1 and MERS-CoV. The amino acid residues which agree with the consensus residues are shown in Figure 1.

### 2.2. Virtual Screening and Molecular Docking

The investigated ligands were obtained from the ZINC natural product database (224,205 ligands). Virtual drug screening was performed to detect the binding ability of the ligands to SARS-CoV-2 nsp16 (PDB: 6W4H). The screening proceeded using high-performance cluster conformant snakemake workflow (paper in preparation). The top 30% of the screened ligands were subjected to rescreening against both SARS-CoV-1 (PDB: 3R24) and MERS-CoV nsp16s (PDB: 5YN6). A total of 6409 ligands showed binding energy less than −8 kcal/mol to all investigated coronavirus nsp16s (Figure 2). The selection criteria of the candidates were based on the binding energy to SARS-CoV-2 nsp16 (compounds that showed the lowest binding energy were selected) and the commercial availability of the compounds.

The chemical structures of the selected ligands and the control S-adenosyl methionine (SAM) are shown in Figure 3.

The selected ligands were subjected to docking using AutoDock 4.2.6 against the three coronavirus nsp16s to visualize the interactions and generate figures. The values of their lowest binding energy (LBE) and their predicted inhibition constants (pKi) are shown in Table 1 along with the values of the natural substrate SAM which was used as a control.

### 2.3. Microscale Thermophoresis

Microscale thermophoresis (MST) was used to detect the binding between hit molecules and the labelled recombinant nsp 16 of SARS-CoV-2, SARS-CoV-1 and MERS-CoV. There was strong binding of SARS-CoV-2 nsp16 to ZINC12880820, ZINC2121012, and ZINC2129028. The K_d_ values of all MST experiments were shown in Table 2, the least K_d_ value achieved by ZINC12880820 was 23.14 ± 1.05 µM.

The three hits were further investigated for binding with nsp16s of SARS-CoV-1 and MERS-CoV. All of them are bound with different K_d_ values, except for ZINC2121012 which is bound to neither SARS-CoV-1 nor MERS-CoV. Representative examples of the binding are shown in Figure 4.

### 2.4. Molecular Docking of Hits to SARS-CoV-2 nsp16 Binding Pocket

Based on the results of microscale thermophoresis, defined molecular docking was performed to assess the binding affinity of the hits to the SARS-CoV-2 nsp16 binding pocket (Figure 5). All candidate molecules share more than one amino acid residue with the control SAM.

Based on the MST findings, both ZINC12880820 and ZINC2129028 were docked against both SARS-CoV-1 nsp16 (PDB: 3R24) and MERS-CoV nsp16 (PDB: 5YN6), the interactions of both hits to the amino acids in the binding pocket are shown in Figure 6 and Figure 7. The results of docking of the hits against nsp16s of SARS-CoV-2, SARS-CoV-2, and MERS-CoV are shown in Table 3.

### 2.5. Cell Viability Assay

The three hits were assayed for their inhibitory effects on the cell viability of human MRC-5 fibroblasts. Only ZINC2129028 showed mild effect with 50% cytotoxicity concentration CC_50_ 18.48 ± 0.66 µM. The other two hits did not show significant effects at concentrations up to 100 µM (Figure 8).

## 3. Discussion

Nsp16 is crucial for coronavirus replication. The methylation of ribose 2′-O of the first nucleotide in the nascent mRNA Cap-0-RNA to form Cap-1-RNA (^m7^GpppA_m2′-O_-RNA) is executed by the Nsp16/10 heterodimer. This methylation prevents the activation of type I interferon-induced response—the significant contributor to host antiviral responses—by RNA sensors in the cytoplasm. Therefore, the task of nsp16 is to obscure the viral mRNAs from the host immune system and, thereby, escape the host immune system response. The inhibition of nsp16 will consequently enhance the recognition of viral RNA by the host [21,22]. The conservation of nsp16 among coronaviruses qualifies it as a potential target for the discovery of coronaviral pan-inhibitors.

In this work, we identified 9 candidate compounds out of a chemical library of more than 200,000 compounds for their binding to nsp16 of SARS-CoV-2, SARS-CoV-1, or MERS-CoV. The multiple sequence alignments of nsp16 in the three investigated coronaviruses revealed high similarities which is consistent with previous results [23]. These similarities are clearly located in the docking poses of the selected ligands, and the residues in the binding domains in SARS-CoV-2, SARS-CoV-1, or MERS-CoV are shared (Figure 5, Figure 6 and Figure 7). Interestingly, all investigated ligands showed higher binding energies and lower estimated inhibition constants than that of the control compound in both screening and docking processes. Three of the 9 candidates were bound to SARS-CoV-2 nsp16 if assessed using MST. These three candidates were further assessed for their binding to SARS-CoV-1 and MERS-CoV, and two of them showed broad-spectrum inhibitory activity to all three investigated coronaviral nsp16s.

In this study, we investigated a natural product library derived from the ZINC database and identified three molecules inhibiting nsp16. These candidates were 8′-methoxy-7-[(naphthalen-2-yl) methoxy]-2H,2′H-[3,4′-bichromene]-2,2′-dione (ZINC12880820), 7′-[(2-methylnaphthalen-1-yl) methoxy]-2H,2′H-3,4′-bichromene-2,2′-dione (ZINC2121012), and 7′-(naphthalen-1-ylmethoxy)-2H,2′H-3,4′-bichromene-2,2′-dione (ZINC2129028). All three candidates contain chromen-2-one as well as naphthalene in their scaffolds suggesting that they are coumarin derivatives. Coumarins are secondary metabolites of plants, they are naturally found in vegetables, fruits, oils, nuts, wine, tea, and coffee. Coumarins have reported anticoagulant, anticancer, antihypertensive, antiadipogenic, antihyperglycemic antioxidant, antitubercular, antibacterial, antifungal, and antiviral properties [24]. Moreover, both natural and synthetic coumarins have reported anti-SARS-CoV-2 activity [25,26].

## 4. Materials and Methods

### 4.1. Multiple Sequence Alignment

The protein sequence and nsp16 nucleotide for SARS-CoV-2, SARS-CoV-1, and MERS-CoV were downloaded from Protein Data Bank (PDB ID 6W4H, 3R24, and 5YN6, respectively). Multiple sequence alignment was performed using Clustal Omega (EMBL-EBI, Wellcome Genome Campus, Hinxton, Cambridgeshire, UK). The preparation of figures was performed using Jalview 2. 11.2.5 (University of Dundee, Scotland, UK).

### 4.2. Virtual Screening of Ligands

A snakemake workflow was used for virtual screening (paper in preparation) This software applies an automatized step of structural-based screening. This workflow uses Open Babel (version 3.0.0) for ligand energy minimization [27], Biopython for the preparation of the target and Vina LC (version 1.3.0, Lawrence Livermore National Laboratory, Livermore, CA, USA) for docking. ZINC database of natural products (22,4205 substances) was screened against SARS-CoV-2 nsp16 (PDB: 6W4H). The best 30% of results were displayed and rescreened against SARS-CoV-1 and MERS-CoV nsp16s (PDB: 3R24 and 5YN6, respectively).

### 4.3. Recombinant Proteins

The recombinant proteins were purchased from the Medical Sciences Institute, School of Life Sciences, The University of Dundee (Dundee, Scotland). The products codes were GST-NSP16 SARS-CoV-2 (DU66420), GST-NSP16 SARS-CoV-1 (DU75116), and GST-NSP16 MERS-CoV (DU75117), they were used in the experimental part of SARS-CoV-2, SARS-CoV-1, and MERS-CoV, respectively.

### 4.4. Microscale Thermophoresis

The 9 selected candidates were purchased from VITAS-M Chemical Limited (Hong Kong, China). The interaction between the ligands and nsp 16 of SARS-CoV-2 was investigated using microscale thermophoresis (MST). The method was described previously [28,29]. Briefly, Monolith™ NT.115 Protein Labeling Kit BLUE-NHS (NanoTemper Technologies GmbH, Munich, Germany) was used for labelling nsp16 of SARS-CoV-2. Then, 16 serial dilutions ranging from 300 μM to 0.0091 μM of each ligand were prepared in the assay buffer (50 mM Tris buffer, pH 7.6 containing 150 mM NaCl, 10 mM MgCl_2_, and 0.05 % Tween-20). The concentration of the labelled protein was measured using Nanodrop 1000 (Thermo Fisher Scientific, Wilmington, DE, USA), then mixed with ligands (1:1) and incubated for 30 min at room temperature. The final concentration of the proteins was 1044 nM, 400 nM, and 264.3 nM for nsp16 of SARS-CoV-2, SARS-CoV-1, and MERS-CoV, respectively.

The 16 samples were loaded into the capillaries in the NanoTemper Monolith™ NT (NanoTemper Technologies GmbH, Munich, Germany). The laser power was adjusted to 20%, 40% and 60% and the LED (Light Emitting Diodes) power was 40%. MO. Affinity Analysis software was used to analyze the data, generation of the fit curve and calculation of dissociation constant (K_d_) [30]. Ligands that showed binding to nsp16 of SARS-CoV-2 were reinvestigated for their binding ability to SARS-CoV-1 nsp16 and MERS-CoV nsp16.

### 4.5. Molecular Docking

The three ligands that showed binding in MST were docked to nsp16s of SARS-CoV-2, SARS-CoV-1 and MERS-CoV (PDB: 6W4H, 3R24 and 5YN6, respectively). AutoDock 4.2.6. (Center for Computational Structural Biology ccsb, California, CA, USA) was used for molecular docking; the binding pockets were identified using the Proteins Plus server (Universität Hamburg). The dimensions of the grid box for 6W4H were adjusted to 74 Å, 70 Å, and 62 Å. The grid spacing value was set to 0.375 Å. The grid centre coordinates were set to 82.114, 16.782, and 25.395 in the x, y, and z directions. For 3R24 the dimensions of the grid box were adjusted to 80 Å, 74 Å, and 60 Å. The grid spacing value was set to 0.375 Å. The grid centre coordinates were set to 57.517, 63.416, and 66.949 in the x, y, and z directions. For 5YN6 the dimensions of the grid box were adjusted to 76 Å, 72 Å, and 66 Å. The grid spacing value was set to 0.375 Å. The grid centre coordinates were set to 63.077, 87.329, and 14.955 in the x, y, and z directions. Lamarckian Genetic Algorithm with 250 runs was applied with 25,000,000 evaluations each. The interacting amino acids were identified using AutoDockTools. The visualization and imaging were created by Discovery Studio Visualizer V 21.1.0.20298 (Dassault Systemes Biovia Corp, San Diego, CA, USA).

### 4.6. Toxicity of the Ligands to Normal Lung Cells

Human diploid MRC-5 lung fibroblasts were provided by Dr. rer. nat. Sebastian Zahnreich (Department of Radiation Oncology and Radiation Therapy, University Medical Center of the Johannes Gutenberg University, Mainz, Germany). The cells were seeded in 96-well plates (5 × 10^5^ cells/well). The treatment was added after overnight incubation of the cells. Ten concentrations of each of the three compounds were used ranging from 0.3 to 100 µM. Aliquots of 20 µL of 0.01% resazurin (Promega, Mannheim, Germany) were added to each well after 72 h incubation at 37 °C and 5% CO_2_. The plates were read after 4 h incubation in similar conditions. Reading was using an Infinite M200 Pro plate reader (Tecan, Crailsheim, Germany) at 550/590 nm. Cell viability was plotted, and the CC_50_ values were calculated as the mean of three independent experiments each of them was six replicates.

## 5. Conclusions

This study provides additional support that different coronaviruses could be targeted with the same inhibitors. We investigated three molecules that bound to SARS-CoV-2 nsp16, and two of them also bound with this protein from SARS-CoV-1 and MERS-CoV. This study created a platform for developing further novel molecules that target nsp16 as a conservative protein in different human coronaviruses.

## Figures and Tables

**Figure 1 molecules-28-00988-f001:**
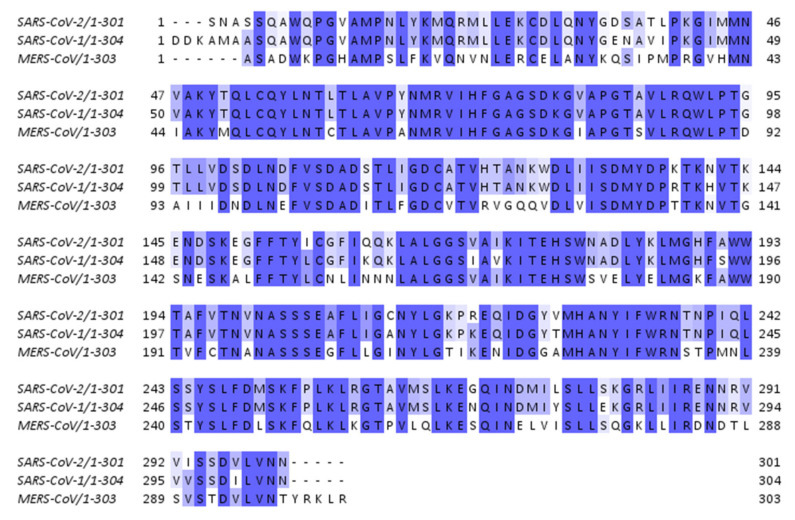
Sequence alignments of nsp16 for SARS-CoV-2, SARS-CoV-1, and MERS-CoV. The dark blue colour indicates that residues show >80% identity with the consensus residues. Blue colour indicates >60%, the light blue colour indicates >40%, and the white colour indicates <40% concordance.

**Figure 2 molecules-28-00988-f002:**
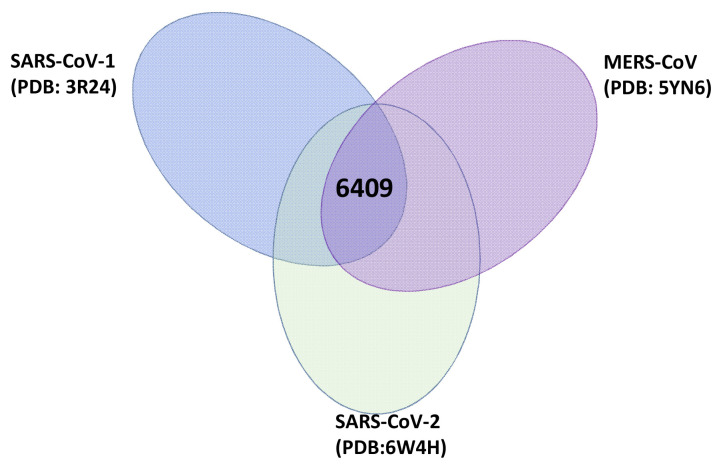
Venn diagram of top 30% ligands bound to SARS-CoV-2, SARS-CoV-1, and MERS-CoV nsp16s.

**Figure 3 molecules-28-00988-f003:**
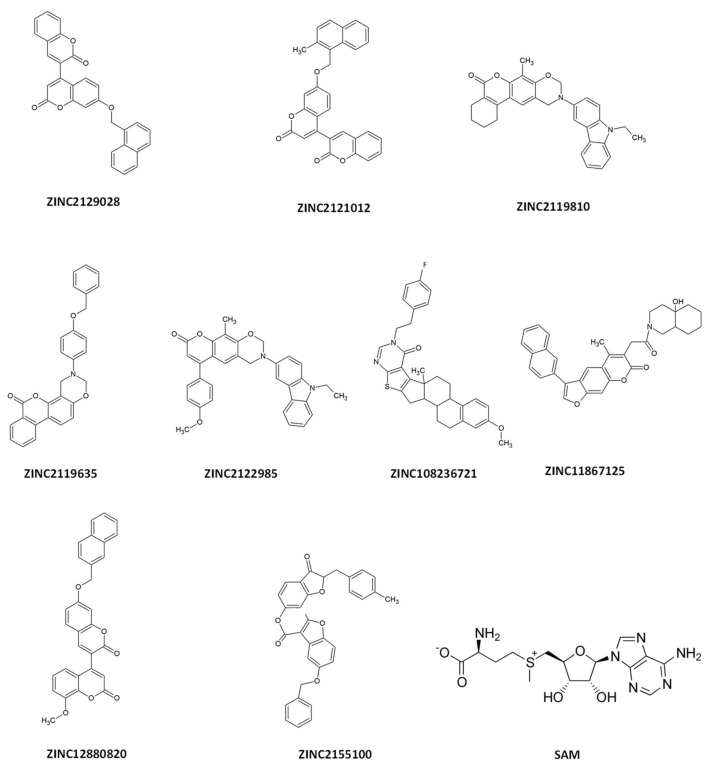
Chemical structures of the selected ligands and SAM.

**Figure 4 molecules-28-00988-f004:**
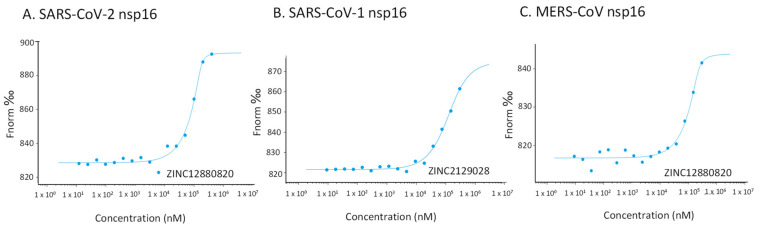
Representative examples of binding of the high-fidelity candidates to nsp16 of SARS-CoV-2 (**A**), SARS-CoV-1 (**B**), and MERS-CoV (**C**) measured by microscale thermophoresis.

**Figure 5 molecules-28-00988-f005:**
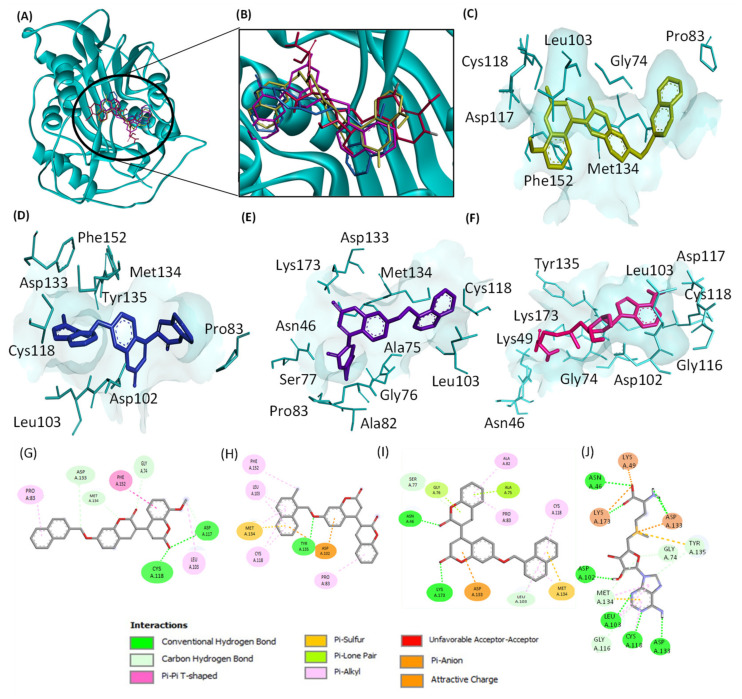
3D and 2D views of the interactions of the hits with SARS-CoV-2 nsp16 binding pocket (PDB: 6W4H) determined by molecular docking. (**A**) Binding of the three hits and the control SAM. (**B**) Zoomed view of the 3D structure of SARS-CoV-2 nsp16. (**C**–**F**): interactions of the amino acids in the binding pocket with ZINC12880820 (gold), ZINC2121012 (blue), ZINC2129028 (purple), and the control SAM (red), respectively. (**G**–**J**) are 2D views of (**C**–**F**), respectively. The images were generated by Discovery Studio Visualizer.

**Figure 6 molecules-28-00988-f006:**
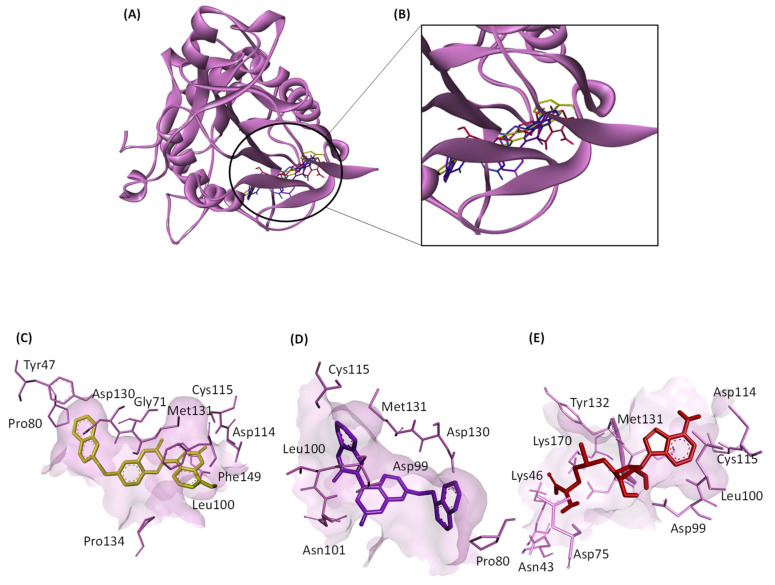
3D and 2D views of the interactions of the hit molecules with SARS-CoV-1 nsp16 binding pocket (PDB: 3R24) determined by molecular docking. (**A**) Binding of the hits and the control SAM. (**B**) Zoomed view of the 3D structure of SARS-CoV-1 nsp16. (**C**–**E**): interactions of the amino acids in the binding pocket with ZINC12880820 (gold), ZINC2129028 (purple), and the control SAM (red), respectively. (**F**–**H**) are 2D views of (**C**–**E**), respectively. The images were generated by Discovery Studio Visualizer.

**Figure 7 molecules-28-00988-f007:**
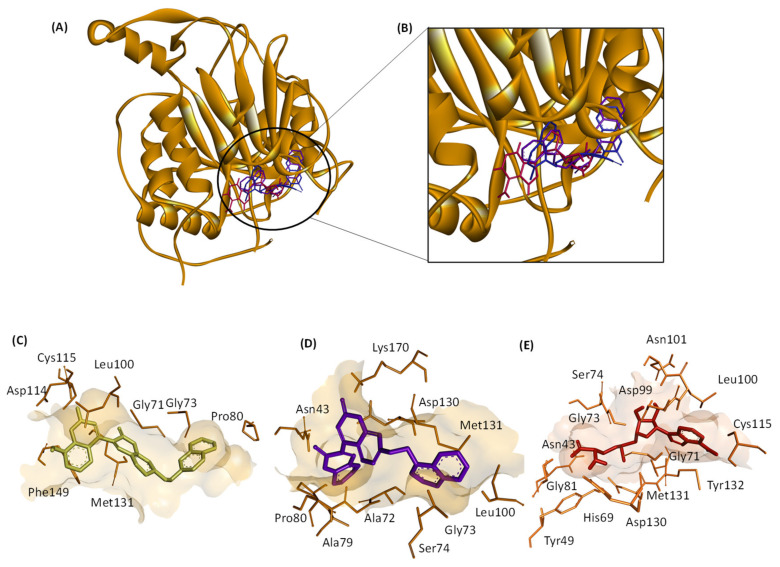
3D and 2D views of the interactions of the hit molecules with MERS-CoV nsp16 binding pocket (PDB: 5YN6) determined by molecular docking. (**A**) Binding of the hits and the control SAM. (**B**) Zoomed view of the 3D structure of MERS-CoV nsp16. (**C**–**E**): interactions of the amino acids in the binding pocket with ZINC12880820 (gold), ZINC2129028 (purple), and the control SAM (red), respectively. (**F**–**H**) are 2D views of (**C**–**E**), respectively. The images were generated by Discovery Studio Visualizer.

**Figure 8 molecules-28-00988-f008:**
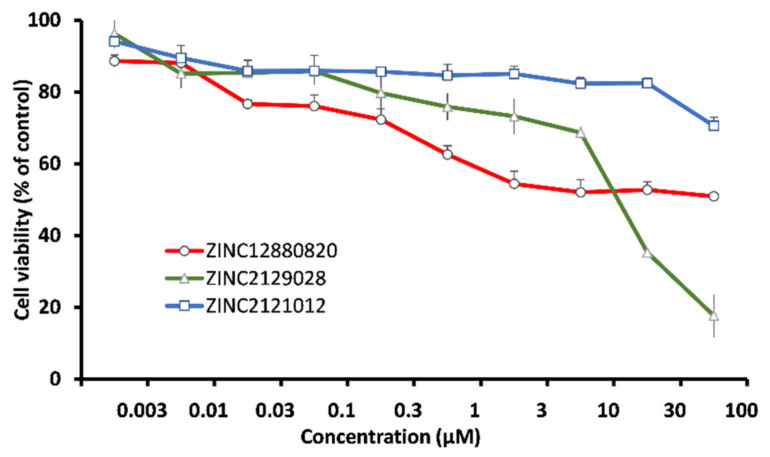
Dose-response curves of the hit compounds against MRC-5 cells as determined by the resazurin assay. The data are representing mean values ± SD of three independent experiments.

**Table 1 molecules-28-00988-t001:** Lowest binding energy of the selected 9 candidates and the control drug SAM upon screening against SARS-CoV-2, SARS-CoV-2, and MERS-CoV. The table also shows the molecular docking results of these candidates against SARS-CoV-2 nsp16 using AutoDock 4.2.6.

Compound	Lowest Binding Energy (kcal/mol) of Compounds Screened against nsp16s			Docking against SARS-CoV-2 nsp16	
	SARS-CoV-2	SARS-CoV-1	MERS-CoV	Lowest Binding Energy (kcal/mol)	Estimated Inhibition Constant (pKi, nM)
ZINC2129028	−13.00	−9.90	−9.10	−13.5 ± 0.48	0.21 ± 0.02
ZINC2121012	−13.00	−10.30	−9.60	−15.48 ± 0.02	4.51 ± 0.15
ZINC2119810	−13.00	−9.80	−9.70	−11.42 ± 0.02	4.30 ± 0.15
ZINC2119635	−13.00	−9.40	−9.40	−12.37 ± 0.00	0.85 ± 0.00
ZINC2122985	−12.90	−10.00	−10.10	−12.25 ± 0.04	1.05 ± 0.07
ZINC8236721	−12.90	−9.10	−9.00	−12.63 ± 0.10	0.56 ± 0.08
ZINC11867125	−12.90	−11.10	−10.40	−13.95 ± 0.00	0.060 ± 0.00
ZINC2155100	−12.80	−10.20	−10.30	−16.11 ± 0.22	0.002 ± 0.00
ZINC12880820	−12.80	−9.30	−10.50	−12.48 ± 0.03	0.70 ± 0.03
SAM	−7.70	−7.10	−6.30	−10.04 ± 0.31	326.93 ± 6.30

**Table 2 molecules-28-00988-t002:** K_d_ values of the binding of the hits to nsp16 of SARS-CoV-2, SARS-CoV-1, and MERS-CoV measured by microscale thermophoresis.

Compound	K_d_ Values (µM) of Candidate Compounds Bound to nsp16s		
	SARS-CoV-2	SARS-CoV-1	MERS-CoV
ZINC12880820	29.08 ± 2.86	>100	36.43 ± 1.12
ZINC2121012	23.14 ± 1.05	-	-
ZINC2129028	59.95 ± 2.53	56.42 ± 1.85	48.11 ± 4.12

**Table 3 molecules-28-00988-t003:** Molecular docking of the hits and the control SAM against nsp16 of SARS-CoV-2, SARS-CoV-2, and MERS-CoV.

Nsp16	Compound	Lowest Binding Energy (LBE, kcal/mol)	Predicted Inhibition Constant (pki, nM)
SARS-CoV-2 (PDB: 6W4H)	ZINC12880820	−12.48 ± 0.03	0.70 ± 0.03
	ZINC2121012	−15.48 ± 0.02	4.51 ± 0.15
	ZINC2129028	−13.5 ± 0.48	0.21 ± 0.02
	SAM	−8.84 ± 0.01	326.93 ± 6.30
SARS-CoV-1 (PDB: 3R24)	ZINC12880820	−13.17 ± <0.0	0.22 ± <0.01
	ZINC2129028	−13.42 ± <0.01	0.14 ± <0.01
	SAM	−8.40 ± 0.07	637.63 ± 8.72
MERS-CoV (PDB: 5YN6)	ZINC12880820	−12.30 ± <0.01	0.97 ± 0.01
	ZINC2129028	−14.08 ± <0.01	0.05 ± <0.01
	SAM	−8.79 ± 0.13	306.05 ± 1.10

## Data Availability

Data is contained within the article.

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
