# Peer review of "Coronavirus Inhibitors Targeting nsp16"

_molecules, 2023, doi:10.3390/molecules28030988_

Round 1

Reviewer 1 Report

Omer et al. performed a computational screen of a natural product based library to identify inhibitors of nsp16. The authors then used generated predicted inhibition constants for the top 9 compounds from the computational screen. The affinity for 3 compounds were determined using microscale thermophresis. Molecular docking was carried out to rationalize the binding mode for the 3 compounds. The 3 compounds were evaluated for the cytotoxicity in human fibroblasts, with 1 of the compounds showing mild toxicity and the other 2 compounds having negligible cytotoxicity. There are some concerns that need to be addressed before the manuscript can be considered for publication.

 Could the authors discuss the rationale for choosing to screen a natural product based library?

 In Figure 1 alignment, does the native SARS-CoV-1 nsp16 sequence have naturally a hexa-histidine tag to it? The front sequence 1-40 residues does not seem to be part of the native sequence. Are there 3 shades of blue (dark blue, blue and light blue)? I could discern only 2 shades of blue.

 Please show MST concentration-reponse graph for non-binding of ZINC2121012 to SARS-CoV-1 and MERS-CoV.

 In Methods section for MST assay, what is the final protein concentration used in MST assay?

 Experimental details need to be given regarding source of recombinant nsp16 from the various viruses in MST assay. Give details regarding recombinant expression and purification of nsp16 if applicable.

Author Response

Reviewer #1:

Omer et al. performed a computational screen of a natural product based library to identify inhibitors of nsp16. The authors then used generated predicted inhibition constants for the top 9 compounds from the computational screen. The affinity for 3 compounds were determined using microscale thermophresis. Molecular docking was carried out to rationalize the binding mode for the 3 compounds. The 3 compounds were evaluated for the cytotoxicity in human fibroblasts, with 1 of the compounds showing mild toxicity and the other 2 compounds having negligible cytotoxicity. There are some concerns that need to be addressed before the manuscript can be considered for publication.

Could the authors discuss the rationale for choosing to screen a natural product based library?

RE: The present study aims to discover natural drug candidates utilizing coronaviral nsp16 as potential druggable target. As previously mentioned, nsp16 is well conserved among coronaviruses as well as it is vital for virus replication. Therefore, we targeted nsp16 to identify broad-spectrum anti-coronaviral candidates that inhibit this protein in SARS-CoV-2, SARS-CoV-1, and MERS-CoV. According to several surveys of the National Cancer Institute, USA, natural products and their derivatives represent the major source of drugs (not only for cancer but for all diseases) (Newman and Cragg, 2020). Therefore, we assume that the identification of chemical scaffolds from natural sources is a good starting point for drug development. Once natural product-based lead compounds are at hand, their pharmacological features can be further improved by synthetic or semisynthetic chemical derivatization. A natural product library derived from the ZINC database was selected for screening as one of the largest databases available at an open access databases with commercially available compounds.

Newman DJ, Cragg GM. Natural Products as Sources of New Drugs over the Nearly Four Decades from 01/1981 to 09/2019. J Nat Prod. 2020 Mar 27;83(3):770-803. doi: 10.1021/acs.jnatprod.9b01285. 

In Figure 1 alignment, does the native SARS-CoV-1 nsp16 sequence have naturally a hexa-histidine tag to it? The front sequence 1-40 residues does not seem to be part of the native sequence. Are there 3 shades of blue (dark blue, blue and light blue)? I could discern only 2 shades of blue.

RE: The figure was corrected. The three colors in Figure 1 in the last line of SARS CoV2 starting from residue number 292 V is light blue, residue number 294 S is blue, residue number 297 V is dark blue.

Please show MST concentration-reponse graph for non-binding of ZINC2121012 to SARS-CoV-1 and MERS-CoV.

RE: The required graphs were added to Figure 4.

In Methods section for MST assay, what is the final protein concentration used in MST assay?

RE: The final concentration of the proteins was 1044 nM, 400 nM, and 264.3 nM for nsp 16 of SARS-CoV-2, SARS-CoV-1, and MERS-CoV, respectively.

Experimental details need to be given regarding source of recombinant nsp16 from the various viruses in MST assay. Give details regarding recombinant expression and purification of nsp16 if applicable.

RE:

4.3. Recombinant Proteins

The recombinant proteins were purchased from the Medical Sciences Institute, School of Life Sciences, The University of Dundee, Scotland. The products codes were GST-NSP16 SARS CoV2 (DU66420), GST-NSP16 SARS CoV (DU75116) and GST-NSP16 MERS (DU75117), they were used in the experimental part of SARS CoV-2, SARS CoV-1 and MERS CoV, respectively.

Reviewer 2 Report

The manuscript “Coronavirus Inhibitors Targeting nsp16” identifies three coumarine derivatives as potentially active compounds against SARS-CoV-2 snp16s. Although the topic of the paper is of high interest, several issues must be addressed before the manuscript could be considered for publication:

- The methodology is not clear, please add supplementary information. For example, how were the top 30% compounds selected? Was used same cut-off  (-8kcal/mol) as for SARS-CoV-1 and MERS1? or the first 30% compounds with the best score were extracted?

- What were the criteria for reducing the number of compounds, from approximately 70,000 (top 30%) to 9 compounds?  If only the binding energy for Sars-CoV-2 was used, what would be the significance of the the other two screenings?  How many compounds had the binding energy below -8kcal/mol?  It is interesting to see if there were 20 compounds and 9 could be ordered or if there were 300 compounds and 9 were ordered….

What were the screening and docking conditions?  How was the binding site identified?  What was the grid size and where was it located?

Why was sinefungin chosen as a control?  For example, SAM- the crystallized ligand - seems a more  suitable control.

Please explain the purpose of docking with Autodock after screening with VinaLC.  (Using the same compounds).
Likewise, the significance of the correlation between the predicted affinity and the binding energy obtained with Autodock.  Maximum correlation is expected…
What is the correlation between the predicted affinity and the one determined exp through MTS?

Author Response

Reviewer #2:

The manuscript “Coronavirus Inhibitors Targeting nsp16” identifies three coumarine derivatives as potentially active compounds against SARS-CoV-2 snp16s. Although the topic of the paper is of high interest, several issues must be addressed before the manuscript could be considered for publication:

- The methodology is not clear, please add supplementary information. For example, how were the top 30% compounds selected? Was used same cut-off  (-8kcal/mol) as for SARS-CoV-1 and MERS1? or the first 30% compounds with the best score were extracted?

RE: The investigated ligands were obtained from the ZINC natural product database (224,205 ligands). Virtual drug screening was performed to detect the binding ability of the ligands to SARS-CoV-2 nsp 16 (PDB: 6W4H). The screening proceeded using high performance cluster conformant snakemake workflow [paper in preparation]. Top 30% of the screened ligands were subjected to rescreening against both SARS-CoV-1 (PDB: 3R24) and MERS-CoV nsp16s (PDB: 5YN6). A total of 6409 ligands showed binding energy less than -8 kcal/mol to all investigated coronavirus nsp16s Figure 2.  (Venn diagram was added)

- What were the criteria for reducing the number of compounds, from approximately 70,000 (top 30%) to 9 compounds?  If only the binding energy for Sars-CoV-2 was used, what would be the significance of the the other two screenings?  How many compounds had the binding energy below -8kcal/mol?  It is interesting to see if there were 20 compounds and 9 could be ordered or if there were 300 compounds and 9 were ordered….

RE: The selection criteria of the candidates were based on the binding energy to SARS-CoV-2 nsp16 (compounds that showed lowest binding energy were selected) and the commercial availability of the compounds. 

What were the screening and docking conditions?  How was the binding site identified?  What was the grid size and where was it located? ).

RE: The three ligands that showed binding in MST were docked to nsp 16s of SARS-CoV-2, SARS-CoV-1 and MERS-CoV (PDB: 6W4H, 3R24 and 5YN6, respectively). AutoDock 4.2.6. was used for molecular docking; the binding pockets were identified using Proteins Plus server (University of Hamburg).

The dimensions of the grid box for 6W4H were adjusted to 74 Å, 70 Å, 62 Å. The grid spacing value was set to 0.375 Å. The grid center coordinates were set to 82.114, 16.782, 25.395 in x, y and z directions. For 3R24 the dimensions of the grid box were adjusted to 80 Å, 74 Å, 60 Å. The grid spacing value was set to 0.375 Å. The grid center coordinates were set to 57.517, 63.416, 66.949 in x, y and z directions. For 5YN6 the dimensions of the grid box were adjusted to 76 Å, 72 Å, 66 Å. The grid spacing value was set to 0.375 Å. The grid center coordinates were set to 63.077, 87.329, 149.55 in x, y and z directions.

Why was sinefungin chosen as a control?  For example, SAM- the crystallized ligand - seems a more  suitable control.

RE: Sinefungin was replaced with SAM. Accordingly, Figures 3, 5, 6, and 7 were edited.

Please explain the purpose of docking with Autodock after screening with VinaLC.  (Using the same compounds).

RE: VinaLC was used as a fast screening program to identify possible candidates out of a large entire chemical library. AutoDock was used as a second independent program to verify the data obtained from VinaLC. Furthermore, AutoDock can visualize the interactions to generate figures. This is not possible with VinaLC.

Likewise, the significance of the correlation between the predicted affinity and the binding energy obtained with Autodock.  Maximum correlation is expected…
What is the correlation between the predicted affinity and the one determined exp through MTS?

RE: We have three data points (each one for SARS-CoV-2, SARS-CoV-1, MERS-CoV). This is too less to calculate statistics. We are sorry.  

Round 2

Reviewer 1 Report

Changes accepted